# Peer review of "Development and Application of a Backscatter Lidar Forward Operator for Quantitative Validation of Aerosol Dispersion Models and Future Data Assimilation"

_Atmospheric Measurement Techniques, 2017_

## Referee Comment (RC1) · Anonymous Referee #1 · 8 Jun 2017

**General comments:**

The study introduces a lidar forward operator in order to simulate the expected lidar observations corresponding to output from a dispersion model such as COSMO-ART, allowing for more direct comparison between the model output and lidar observations from instruments such as CALIOP or automated ceilometer lidar systems. Rather than using a fixed lidar ratio, the new forward operator calculates the scattering properties of the particle mixture specified by the dispersion model. The COSMO-ART simulation of the 2010 Eyjafjallajökull eruption is used as a case study.

The computationally efficient method for calculating the aerosol scattering properties, backed up by sensitivity studies, and the flexibility of the resulting model make this work an important contribution. The writing style is very clear, but could be more succinct: some of the background information seems unnecessarily detailed and detracts from the focus of the study.

Specific comments:

Page 2, lines 1-24. This discussion of different dispersion models and different lidar configurations doesn't come up much later in the manuscript. Is it possible to tie the most relevant aspects more closely to the focus of the study, and omit the rest?

Page 3, lines 1-12. Does each of these lidar forward operators simulate CALIOP, a ground-based lidar network, or both?

Pages 4-11. The equations that are not new to this study should have sources cited; the better-known aspects of lidar physics can probably be described more briefly.

Page 8, lines 2-7. Likewise, the substitution of sums for integrals in the numerical computation is straightforward enough that I'm not sure we need the steps spelled out explicitly.

Page 10, line 11. For Fig. 1, can you overlay the track of the CALIPSO overpass you show in Fig. 3? It would be helpful to connect the overhead and profile views of the ash plume.

Page 18, lines 1-6. The "missing" ambient boundary-layer aerosol seems as if it would also make quantitative comparison very difficult.

Technical comments:

Page 1, line 20. By "aerosol cloud movements" do you mean the movement of aerosol and clouds, or of aerosol plumes?

Page 5, lines 3-5. This sentence is unclear.
Page 7, line 1. Should be "Mie scattering-related"

Page 10, line 10. "Spatial extent"

Page 17, line 1. "Predominantly"

Page 17, line 4. Should be "This may be important information"

Page 20, line 16. "A comparison of the volcanic ash signal"; "was too high"

---

## Referee Comment (RC2) · Anonymous Referee #2 · 10 Jul 2017

General comments

The paper presents a new backscatter-lidar forward operator based on the distinct calculation of the aerosol backscatter and extinction properties. This operator was then applied to a specific case study: the Eyjafjallajökull eruption in 2010. It is noted the importance of developing a forward operator based not on a fixed lidar ratio. This approach is really valuable and could have significant impact on future data assimilations schemes. The paper is generally clearly written and the topic is suitable for AMT. However there are still aspects to be better exploited before the paper can be accepted for

publication on AMT.

I understand that the chosen case study is considered suitable because in case of a volcanic eruption the source is known and very limited in space and time. However it is well known that volcanic ash particles are not spherical and this increase the complexity of the case study. The authors have addressed this issue using, in a second step, the T-matrix approach for sensitivity studies and they consider this approach sufficient. Probably this statement should be better analysed. In particular, I'd suggest to reduce as much as possible the other error sources. Limitation for particles larger than 10 micron can not be reduced (see also Madonna et al. JGR 2013, 10.1002/jgrd.50789), because it is intrinsic in the lidar technique (even if with larger wavelength, as in the case of ceilometers, the sensitivity improves but it is not sufficient for this particle size range). On the other side calibration errors and eventual presence of aerosol background could be reduced using data from a Raman lidar or HSRL.

I understand that the Eyjafjallajökull eruption in 2010 was a great opportunity to present the potential of the German ACL network, however in this paper only data from a single ceilometer station are used. I do not see the need to present the network here. Actually, even if I understand the interest of using this kind of ceilometers instead of more advanced research lidar (clearly explained in the paper), I do not understand why the comparison with the forward operator is not done first with a more advanced lidar (at least measuring independent extinction and backascatter profile) [Raman lidar o HSRL]. It is not necessary to use a multiwavelength Raman lidar if the authors do not want to, but at least a lidar providing independent extinction and backscatter profiles. This could have helped the assessment of the operator reducing calibration uncertainty and also improving signal to noise ratio at altitude where is more realistic to neglect aerosol background and consider only volcanic particles.

The authors could then perform a sensitivity test in case of ceilometers. I have to say that the smart idea of developing this operator does not fit with the calibration of the ACL with CALIPSO (which is very weak). I suggest a more quantitative approach first

and then to try to apply it to simpler lidar as the ceilometers, which I agree we should better investigate the specific potential because even if more Raman lidar and HSRL are becoming more "operational" the coverage will be never comparable as that we have for the ceilometers.

Specific comments

Probably the reference Matthias et al., atm envir 2011 should be cited. Extinction profiles simulated with a fixed lidar ratio.

Fig. 2 reports the distribution of the German ACL network: it is a pity that only 6 stations are reported in green (clean air situation with a full view on the ash layers ). This is not commented at all in the text. It should be probably explained or at least mentioned what are the limitation of the use of these instruments (it is only reported that they are more operational respect to research lidar). However as I said previously, I do not see why this network should be presented considering that it is not used in the paper.

Calibration of the ACL with CALIPSO is very weak: no ideal co-location, limited signal to noise ratio for the ACL.

Comparison between ACL and the model has no sense in the PBL because only volcanic particles have been considered and here it is not possible to neglect other aerosols (not only in the PBL, looking at the fig 15 and 16 at least up to 2 km of altitude). The ACL signal is too noisy between 4 and 8 km of altitude where a much more realistic comparison with the model could have been done considering only volcanic particles. It is unrealistic to calibrate at 3 km of altitude supposing only molecular contribution at this altitude in Central Europe (Pag 11, line 16).

The comparison with a much better calibrated lidar is recommended. There should be many Raman or HSRL data available for the considered case study which could be used for this comparison. To use better calibrated lidar data would have allowed a better quantitative comparison, as also mentioned by the authors. I strongly recommend

the authors to use this kind of data for the comparison with the model. Then, once the operator is better assessed could be exported to the specific case of ceilometers.

Pag 3 line 15, extinction less sensitive to dimension and shape, but not on refractive index

Pag. 10, lines 1-3: there are many lidar ratio data available for this case. Please add appropriate references. The authors state here "missing reference measurement data" which is in contrast with what reported at pag. 19, line 31.

Pag 20 lidar ratio di 5sr?? this value is completely out of the range of values observed in case of volcanic ash.

Figures are not generally sufficiently commented.

References to relevant papers and important programmes in the field are omitted. Considering ceilometers networks, it is suggested to mention E-Profile at least in the conclusion.

Sections 2-4 could be more succinct.

Technical comments:

- The acronym "ACL" is reported in the abstract where it is not explicitly explained

- Page 17, line 1. "Predominantly"

- Pag. 20, line 16: A comparison "with" the volcanic . . .

- Please check the quality/resolution for fig. 2

- Figures 5 and 6: The readability of the text should be improved

- Figures 7-8-9-10: axis labels should be enlarged

---

## Author Comment (AC1) · 14 Sep 2017

We thank the two reviewers for reading the manuscript so carefully and providing detailed and very valuable comments which helped to further improve the manuscript substantially. We have carefully addressed all comments and changed the manuscript accordingly.

The final author comments with responses to all Referees suggestions and the revised manuscript with changes highlighted is attached as supplement to this comment.

[Figure]

Please also note the supplement to this comment:
https://www.atmos-meas-tech-discuss.net/amt-2017-142/amt-2017-142-AC1-
supplement.pdf

―――――――――――――――――――